# Personalizing Communication of Clinicians with Chronically Ill Elders in Digital Encounters—A Patient-Centered View

**DOI:** 10.3390/healthcare12040434

**Published:** 2024-02-08

**Authors:** Gillie Gabay, Hana Ornoy, Attila Gere, Howard Moskowitz

**Affiliations:** 1Faculty of Social Sciences, Achva Academic College, Arugot 7980400, Israel; 2Faculty of Business, Ono Academic College, Kiryat Ono 5545173, Israel; hana.or@ono.ac.il; 3Institute of Food Science and Technology, Hungarian University of Agriculture and Life Sciences, 1118 Budapest, Hungary; gere.attila@uni-mate.hu; 4Mind-Genomics Associates, White Plains, NY 10605, USA; mjihrm@gmail.com

**Keywords:** communication, digital medical encounters, elderly chronically ill patients, experimental design, health expectations, mindsets, patient-centered care

## Abstract

Background: Chronically ill elderly patients are concerned about losing the personal connection with clinicians in digital encounters and clinicians are concerned about missing nonverbal cues that are important for the diagnosis, thus jeopardizing quality of care. Aims: This study validated the expectations and preferences of chronically ill elderly patients regarding specific communication messages for communication with clinicians in telemedicine. Methods: The sample comprised 600 elderly chronically ill patients who use telehealth. We used a conjoint-based experimental design to test numerous messages. The outcome variable is elder patient expectations from communication with clinicians in telemedicine. The independent variables were known categories of patient–clinician communication. Respondents rated each of the 24 vignettes of messages. Results: Mathematical clustering yielded three mindsets, with statistically significant differences among them. Members of mindset 1 were most concerned with non-verbal communication, members of mindset 2 prefer communication that enhances the internal locus of control, and members of mindset 3 have an external locus of control and strongly oppose any dialogue about their expectations from communication. Conclusions: The use of the predictive algorithm that we developed enables clinicians to identify the belonging of each chronically ill elderly patient in the clinic to a sample mindset, and to accordingly personalize the communication in the digital encounters while structuring the encounter with greater specificity, therefore enhancing patient-centered care.

## 1. Introduction

The Institute of Medicine defines patient-centered care (PCC) as respectful care that responds to the preferences, needs, and values of each patient [1,2]. Evidence suggests that PCC results in higher patient trust, lower costs of care, shorter hospital stays, higher patient self-efficacy, better self-management of illness, lower stress among clinicians, higher adherence, lower re-admissions to the hospital, improved clinical outcomes, and higher patient satisfaction [2,3,4,5,6]. PCC entails the beliefs and needs of the patient and a partnership is established between patients and clinicians [7,8,9]. Studies on PCC highlight the formation of personhood as an activity that requires interaction with others so that the clinician as a partner has a crucial role in supporting personal communication with vulnerable patients [10,11,12]. This approach to care calls for a therapeutic alliance that encompasses effective patient–provider communication [13]. Although PCC is the advocated approach to care, health systems are still far from obtaining it [14]. Furthermore, the degree to which PCC is delivered at the clinician level and the individual patient level is not characterized [1,9]. Health systems have established platforms for digital medical encounters (telemedicine) for patients, including for chronically ill elders. 

Telemedicine, i.e., providing a wide range of medical care and information from a distance mediated by technology, has been globally expanding since the outbreak of the COVID-19 pandemic [15,16,17,18]. Telemedicine is being integrated into practice in value-based health systems, increasing accessibility to health services and delivery of care [19,20,21]. Telemedicine visits conveniently connect clinicians to patients in their own homes as well as connect clinicians to each other, enhancing coordination of care [22]. Health systems aspire to utilize telemedicine platforms across 51 chronic illnesses and patients living in rural distant cities [23,24,25,26]. However, telemedicine alters the face-to-face encounter, negatively affecting personalization in patient–provider communication [27]. Adopting the lack of face-to-face interaction that characterizes telemedicine may fail to satisfy the emotional needs of elderly patients while also decreasing their engagement [28]. 

Patient-tailored communication in telemedicine was found to improve clinical outcomes, but barriers of depersonalization decrease patient satisfaction and involvement in telemedicine, jeopardizing effective communication [29,30,31]. Also, with elders, communication itself is a barrier to innovative mobile health solutions [32]. As for clinicians, they feel that without the personal encounter, assessing patients’ emotional states and attending to their needs as individuals is challenging [31]. Clinicians acknowledged that their assumptions about their patients’ needs may be mistaken, impairing quality care [29]. Although both patients and clinicians articulate some unease about the lack of face-to-face interaction, clinicians stress the importance of understanding their patients’ expectations and preferences [29]. While patients focused on issues of non-verbal communication and the lack of personal “connection”, clinicians focused on the potential for “missing something” medically important when delivering care to elder patients [29]. 

The dearth of original research on telemedicine in primary care suggests limited evidence on utility and quality [33]. Provider–patient communication in telemedicine must enable chronically ill patients and clinicians to gain a higher sense of control in the encounter while supporting a holistic view of care [19,34,35]. Previous studies tested provider–patient interaction frequency [36], interactional unfairness [37], communication competencies [38], and engagement in the interaction [39]. One study demonstrated that telemedicine may be adequate for the empathetic connection of clinicians with patients, allaying patients’ concerns, offering emotional and informational support, and enhancing patient cooperation [40]. Furthermore, multiple studies tested self-disclosure, social support, and patient–provider trust in telemedicine [41,42,43].

Based on the PCC view, patient-tailored communication may enhance the engagement of elderly chronically ill patients in telemedicine encounters, contributing to clinical outcomes [30,44,45]. Research on specific communication messages that patients prefer in communication with clinicians, which may guide them to better customize telemedicine communication with elderly chronically ill patients, is scant. More insight is essential to remove current barriers among elderly patients in telemedicine [46]. Previous studies called to illuminate how clinicians may personalize communication in telemedicine to provide elderly chronically ill patients with PCC and optimize the quality of care in telemedicine [29,30,33,43,46,47]. This study validates the expectations and preferences of chronically ill elderly patients regarding specific communication messages for communication with clinicians in telemedicine, testing the following questions: What messages do elderly chronically ill patients prefer in communication in telemedicine? Do different subgroups of chronically ill patients differ in their preferences for telemedicine communication with clinicians? To the best of our knowledge, this is the first study to validate communication preferences in the elderly. 

## 2. Materials and Methods

### 2.1. Ethics

The institutional board of ethics at the academic college in which the second author is affiliated granted ethical approval for this study, in accordance with the 1964 Declaration of Helsinki.

### 2.2. Sample

The sample comprised 600 chronically ill elderly patients who received medical care through digital encounters since the outbreak of the COVID-19 pandemic and accords with the suggested number of respondents in studies that aim at stable utilities [48]. The inclusion criteria were chronic illnesses such as cardiovascular diseases, neurological diseases, diabetes, metabolic disorders, lung disease, chronic back pain, and osteoporosis. Exclusion criteria were mental illness and cancer, which differ in their constructs. Six hundred undergraduate students in business administration programs at one of four campuses of the academic institution with which the second author is affiliated interviewed patients. A total of 90 interviewers are from the first campus, 100 interviewers are from a second campus, 350 interviewers are from the third campus, and 60 interviewers are from a fourth campus. The data were obtained as part of the students’ research obligations in a research laboratory course. Each student was asked to interview one elderly chronically ill patient in the age group of 65 and over from a nursing home. The students were trained by the second author. They read the questions to the patient and entered their responses on the online questionnaire. This study was active between 15 June 2021–16 November 2021 as part of a series of testing expectations of patients from communication in telemedicine. 

### 2.3. Procedure

This study is part of a series. The first study in this series focused on the expectations of young patients from doctors in telemedicine [17]. This study focuses on the expectations of chronically ill elders from communication with clinicians in telemedicine. For all papers in this series, we applied a conjoint-based experimental design. Typical of such designs, participants are allocated into groups which all respond to repeated measures, under the same conditions [49]. The respondents were asked to rate vignettes of messages relating to their expectations from communication with clinicians in telehealth. Using a computerized program, the order of the vignettes was alternated. Compared with observational studies, this alteration reduced potential bias by increasing the variation, randomization, co-variance analysis, and control [48]. Simulating the complexity in reality, where multiple stimuli may operate together, we tested numerous communication messages with no limitation of degrees of freedom through the Mind Genomics conjoint experimental design methodology [50]. Responses to the vignettes of messages were statistically analyzed [48,50,51]. Conjoint-based experimental designs require the setting of categories and elements. We set four categories determined by a thorough literature review to identify independent variables and four elements to be used as communication messages under each category representing patient expectations regarding communication with clinicians [50,51]. Each respondent evaluated a unique set of 24 combinations of messages, constituting a full experimental design [51].

### 2.4. Instrument

After reading the orientation page of the study and signing the informed consent for participation and publication, respondents answered three demographic questions and consequently rated 24 vignettes of messages. The outcome variable was elders’ expectations of communication with clinicians in telemedicine. The rating question was: “How important are these communication messages when communicating with clinicians in telemedicine?” The rating scale ranges from 1 (not at all important to me) to 9 (extremely important to me). The independent variables were four acknowledged patient expectations of clinician–patient communication: locus of control [44], education and health literacy [52,53], and attentive listening [17]. The sixteen messages were arranged into 24 vignettes, with each category consisting of two to four messages, for a total of 24 vignettes that each respondent rated. The respondents were asked to rate the vignettes as a unit rather than relate to each message separately [50,51]. 

A conjoint-based software organized the random order of the vignettes ensuring statistical independence of the independent variables for subsequent regression analyses at both the individual level and the group level [50,51]. Table 1 presents categories and the messages regarding expectations in each.

### 2.5. Data Analysis

Ordinary least regression was performed on an individual level for each message [50]. Ordinary least regression was performed for the whole sample by gender and age group to test the contribution of each message to the importance of the vignette of messages. Next, k-means clustering analysis was performed to uncover the response patterns of respondents to each message with 1-Pearsons’s R distance measure [48,50,51]. This group-level analysis of response patterns yielded subgroups (mindsets). ANOVA and Tukey post hoc tests were performed to test if differences among mindsets were significant [52,53,54]. The high coefficients in each mindset guided the characterization of what is most important to members of each mindset. To connect the insights from this study to practice, we developed a predictive algorithm to easily assign a chronically ill elder patient in the population to a sample mindset.

## 3. Results

### 3.1. Preliminary Analysis

We used split halves to test reliability by testing internal consistency between the data for the whole sample and the data for half of the sample (0.70; 0.76). Sixty-one percent were females (367), and thirty-nine percent (233) were males. The age ranged from 65 to 92 years. The age distribution amongst the four age groups was as follows: 377 respondents, age group 65–70; 140 respondents, age group 71–79; 59 respondents, age group 80–87; and 24 respondents, age group 88–95.

### 3.2. Secondary Analysis

Data analysis was conducted using R-project (version R-3.6.0) and lm.beta package (R Core Team, Vienna, Austria, 2020). The additive constant is the intercept in the linear regression equation and presents the likelihood that respondents will agree with a vignette of messages. High additive constants (60+) represent groups who strongly agreed with the vignettes and low additive constants (<35) represent groups whose agreement with the vignettes was low. The additive constant was 0.53 for the whole sample and 0.55 for the age group of 65–70, reflecting the high importance of the subject. To crystalize the results of the importance of each message, this outcome variable was recoded as a categorical variable. Ratings of 1–6 were recoded as zero and ratings of 7–9 were recoded as 100. 

The regression analysis yielded coefficients for the 16 messages. The higher the coefficients, the higher the importance of the message. In conjoint analysis, coefficients of 6 and higher are statistically significant indicating a high importance of the message. Coefficients that are lower than 6, or negative coefficients, indicate a lack of agreement with the message. The regression analysis was performed for the whole sample by gender and age testing the contribution of each message to the importance of the whole vignette of messages [50]. With 600 respondents and 24 vignettes rated by each respondent, there were 600 (14,000) different observations for the regression analysis (Systat Software Inc., San Jose, CA, USA, SYSTAT v13.2. 2020. www.sigmaplot.com). Regression coefficients for the total sample were not significant, indicating that there were no significant differences in the importance of messages in communication via telemedicine. 

Likewise, no significant differences were found between the coefficients of males and females across age groups. Looking into differences among age groups, the strongest expectation of respondents in the age group of 80–87 was “Upon entry to the telehealth visit, the clinician greets me warmly” (beta = 10). The strongest expectations of respondents in the age groups of 88–95 were: “The clinician provides me with specific information for my conditions, not just general info” (beta = 11) and "The clinician walks me through a process of change to maintain my health" (beta = 12). 

Next, using k-means mathematical clustering, we analyzed response patterns to each message with 1-Pearsons’s R distance measure [48,50,51]. Commonalities in response patterns to each message uncovered groups of participants with similar responses to messages [52]. The clustering analysis yielded three subgroups (mindsets). ANOVA and Tukey post hoc tests showed indices that indicated that the differences among mindsets were significant [54]. The three mindsets are based on mathematical clustering, from which the commonalities in patterns of response to each message facilitated the assignment of each participant into mindsets presenting distinct expectations of elder patients from communication with clinicians in telemedicine. The mindsets reflect various ways of thinking about communication with clinicians. In each mindset, the highest coefficients guided the characterization of the expectations of members in that mindset [55,56]. Table 2 presents mindset segmentation. Figure 1 illustrates the mathematical clustering of messages by importance.

Members of mindset 1, comprising 36% of the sample, were most concerned with non-verbal communication; they sought a warm greeting and eye contact. The growing elderly population and the immense benefits of telemedicine call upon clinicians to address the needs of chronically ill patients in telemedicine communication and optimize its utilization [55]. Members of mindset 2, comprising 32% of the sample, are interested in communication that enhances internal locus of control. They would like to talk about their expectations and for the clinician to guide them through a change process to take responsibility and maintain their health. They would like relevant specific information about their condition rather than general information about the disease. The strongest messages for this mindset were “The clinician walks me through a change process to maintain my health” (beta = 12); “The clinician talks to me about my expectations” (beta = 12); and “The clinician helps me take responsibility for my health” (beta = 9). They expect clinicians to use messages that empower and navigate them to better manage their illness leading to higher resilience [57,58,59]. Members of mindset 3, comprising 31% of the sample, have an external locus of control and strongly oppose a change process guided by the clinician to maintain their health (beta = −12), to help them take responsibility for their health (beta = −13), or dialogue about their communication expectations (beta = −14).

To connect the insights from this study to practice, we developed a predictive algorithm. To develop the algorithm, we considered a total of sixty-four potential response patterns to each message and the messages that strongly distinguished among the three mindsets. Based on six messages that distinguished among the mindsets, we employed a mathematical Monte Carlo simulation and developed the predictive algorithm [55]. The predictive algorithm enables clinicians to easily assign a chronically ill elder patient in the clinic to one of the study mindsets and choose to communicate by mindset-tailored communication messages. Since respondents are distributed across age groups and genders in the population, we created a predictive algorithm to quickly assign elderly patients to mindsets and accord communication to meet the expectations of members of each mindset. Figure 2 presents the predictive algorithm available at https://www.pvi360.com/TypingToolPage.aspx?projectid=2312&userid=2008 (accessed on 19 July 2022).

## 4. Discussion

This study tested expectations of elderly chronically ill patients based on communication messages in telemedicine, presenting a novel strategy of testing patterns of thinking and expectations of elders from communication in telemedicine, creating a predictive algorithm for detecting patient mindset and using mindset-tailored communication with clinicians. The identified mindsets contradict previous findings, which viewed patient expectations from telehealth services as varying by services provided and respondent demographics [60]. This study filled several gaps in the knowledge. First, this study identified what elders with chronic illness expect and perceive as important in communication with clinicians in telemedicine and how they are segmented in their expectations. Methodologically, this study conducted a conjoint-based experimental design that reduces biases of typical observational studies while testing a great many messages with no limitations of degrees of freedom. 

This work allowed us to answer the following research questions:(a)What messages do elderly chronically ill patients prefer in communication in telemedicine? Surprisingly, no significant differences were found between male and female chronically ill elders across age groups. The strongest expectation of elders ages 80–87 was that the clinicians will greet them warmly upon the beginning of the virtual encounter. The strongest expectations of elders ages 88–95 were that the clinicians will provide them with information that is specific to their conditions, rather than general information, and that clinicians will guide them through a process to maintain their health.(b)Do different subgroups of chronically ill patients differ in their preferences for telemedicine communication with clinicians? Three mindsets, which are similar in size, represent subgroups of chronically ill patients and the expectations of members of one mindset are significantly distinct from expectations of members of other mindsets. Thus, expectations of members of one mindset are irrelevant to members of other mindsets. Members of mindset 1 expect a warm greeting and eye contact. Members of mindset 2 expect communication that enhances their internal locus of control, a dialogue concerning their expectations, and for clinicians to guide them through a change process to maintain their health. These elders expect specific information about their condition and for the clinician to help them take responsibility for their health, empower them, and navigate them to better manage their illness, leading to higher resilience. Members of mindset 3 have an external locus of control and oppose dialogue and a change process guided by clinicians.

Although the COVID-19 pandemic brought about rapid adaptation to telemedicine to complement “in-person” primary care, it also highlighted challenges [33]. Telemedicine is indeed a key response in primary care to support public health. Patient experience in telemedicine, however, entails barriers for clinicians and patients that decrease utility and quality of care [33]. This study begins to validate the removal of barriers to both patients and clinicians in communication in telemedicine with elderly chronically ill patients [61]. While regression on the total sample indicated no significant differences among chronically ill patients by expectations from communication, mathematical clustering yielded three distinct mindsets. Thus, offerings of communication in telemedicine should not be tailored to different demographic groups but rather to mindsets. These findings echo previous studies that suggested that patients should be segmented by their expectations from clinicians based on their mindset-belonging description rather than on demographics, driving patient satisfaction and health behaviors and enabling clinicians to use mindset-tailored communication to meet patient expectations and enhance the quality of care [57,58,59,62,63,64]. 

The novel strategy we presented to test communication expectations of chronically ill elders in telemedicine is not without limitations. First, the generalizability of this study. While the presented novel statistical strategy itself is replicable and generalizable across health settings and countries and is expected to yield distinct mindsets of patients, communication has cultural roots, which reduces the generalizability of the communication messages based on mindsets. First, attitudes toward the elderly differ between Western and Eastern cultures; in the latter, the elderly are highly respected, and all their needs must be fulfilled affecting patient–clinician communication in telemedicine. Second, there are cultural differences in attitudes and perceptions of authority figures that may affect the involvement of elders in communication with clinicians as authority figures, regardless of whether the clinician will use mindset-tailored communication messages. Third, our global landscape is becoming rich in immigrant diversity and elderly immigrants may not be fully versed in the dominant local language. Another limitation is that we did not test elder patient expectations by diseases included in this study, which may yield different mindset-belonging description results and varying importance levels of the messages in each mindset. Along with the patient-centered care framework, future studies may test the expectations of chronically ill patients from clinicians, by disease and by culture. Also, future studies may test the effect of implementing mindset-tailored messages with chronically ill elder patients on their satisfaction with interactions with clinicians in telemedicine.

## 5. Relevance to Clinical Practice

Telemedicine is developing, expanding, and creating new opportunities to meet patient expectations, improve the quality and delivery of care by personalizing care, and closing the chasm between aspiring for PCC and obtaining PCC. To meet patients’ expectations and to put patients into focus, clinicians are advised to support communication with patients using mindset-tailored messages, which are considered the most important emphases for members of each mindset. The use of the predictive algorithm developed for this study detects the mindset-belonging description of each elder patient at the clinic linking it to a sample mindset, enabling personalized communication while delivering care via telemedicine. Understanding the expectations of patients regarding communication with clinicians based on mindset can structure the telehealth visit more effectively, thus enhancing patient-centered care and quality of care. Moreover, greater personalization will remove a major barrier for patients regarding telemedicine. Since personalized communication was proven to establish higher patient trust in clinicians, it may yield trust in clinicians in telemedicine as well. Trust was proven to enhance medication adherence and self-management of diseases [65]. Hence, it is possible that using mindset-tailored communication with chronically ill elder patients will contribute to optimal self-management of chronic diseases, creating an added value to telemedicine and enhancing the quality of care [66,67,68,69,70,71]. Greater utilization of telemedicine by patients and greater patient involvement may increase if clinicians meet patient expectations in communication with clinicians, which may improve health behaviors and overall clinical outcomes [27,68].

## 6. Conclusions

In terms of practice implications, the insights developed from this study enable clinicians to employ our predictive algorithm, or develop one, to easily assign elder patients with chronic illnesses (excluding mental illness) in telemedicine into mindsets from the sample. Detecting the mindset to which a patient belongs and understanding the strongest expectations of members in that mindset will provide clinicians with the choice to communicate with the patient through personalized, mindset-tailored messages.

## Figures and Tables

**Figure 1 healthcare-12-00434-f001:**
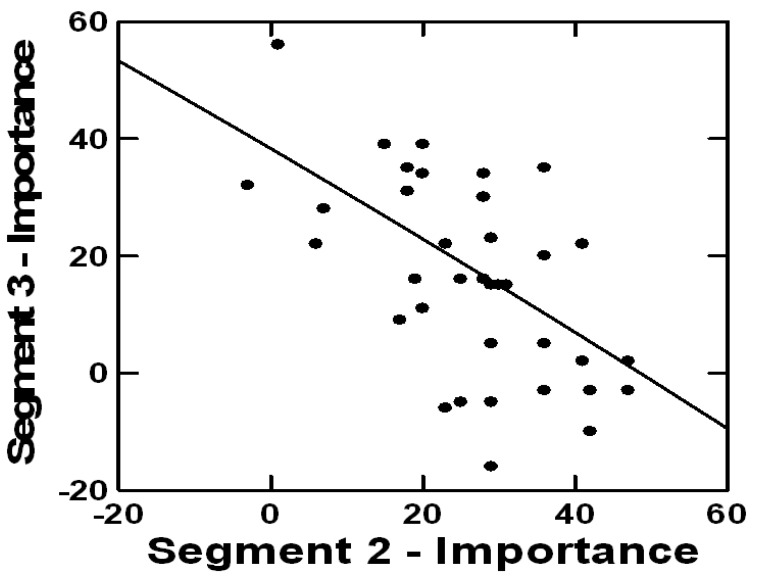
An example of mathematical clustering to identify commonalities by the perceived importance of messages.

**Figure 2 healthcare-12-00434-f002:**
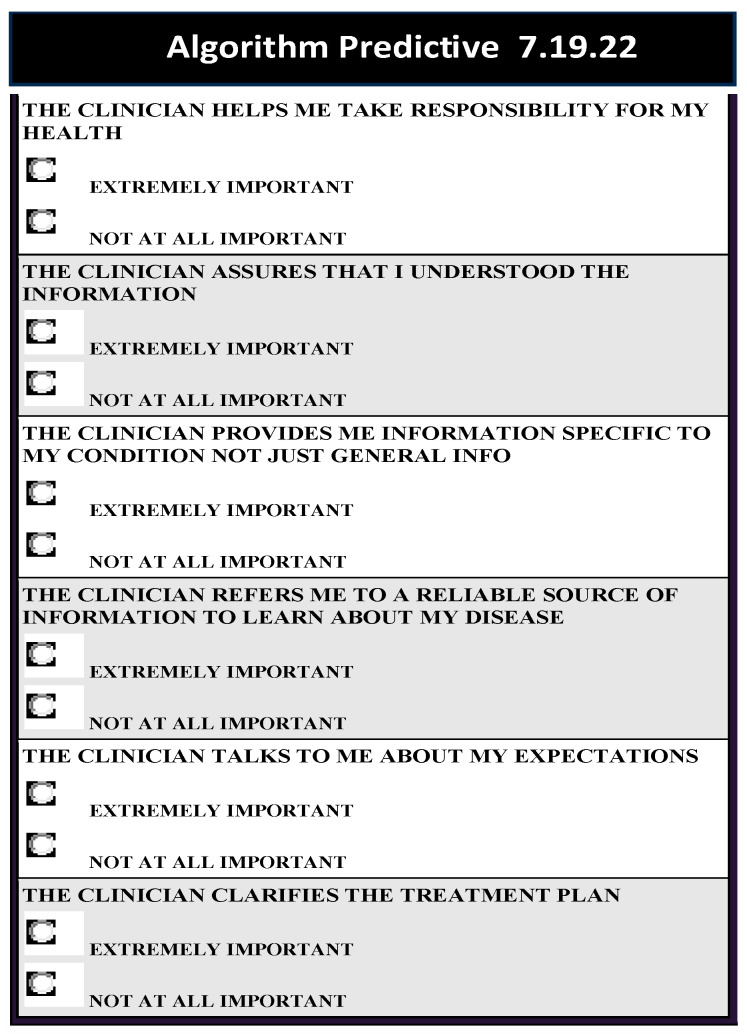
The predictive algorithm listing the six most discriminating elements with binary answers needed to predict the mindset membership of new participants.

**Table 1 healthcare-12-00434-t001:** Categories and the messages regarding expectations in each.

**Category 1—Locus of Control**	
A1	The clinician walks me through a process of change to maintain my health.
A2	The clinician initiates a dialogue about my expectations.
A3	The clinician explains the treatment plan.
A4	The clinician navigates me to take responsibility for my health.
**Category 2—Health Literacy**	
B1	The clinician provides me with specific information for my conditions, not just general info.
B2	The clinician recommends an accredited site for information to learn about my condition.
B3	The clinician ensures that I understood the information.
B4	The clinician encourages me to ask questions.
**Category 3—Attentive Listening**	
C1	The clinician invests all the time that I need.
C2	The clinician cares about my feelings.
C3	The clinician enables me to express myself without interrupting me.
C4	The clinician invests more time if I need it.
**Category 4—Non-verbal Communication**	
D1	The clinician respects my time.
D2	The clinician does not maintain direct eye contact with me.
D3	The clinician is short and succinct and even sends me the prescription.
D4	Upon entry to the telehealth visit, the clinician greets me warmly.

**Table 2 healthcare-12-00434-t002:** The results of segmentation by mindset (MS) and Tukey post hoc tests.

	Group (Binary Ratings) Total		MS1	MS2	MS3
	**Base Size**	600	217	194	189
	**Additive Constant**	53	54	53	51
	**Category A: Internal Locus of Control**				
A1	The clinician walks me through a process of change to maintain my health.	1	6 ^b^	**9 ^b^**	**−12 ^a^**
A2	The clinician initiates a dialogue about my expectations.	1	6 ^b^	**12 ^b^**	**−14 ^a^**
A3	The clinician explains the treatment plan.	1	7 ^b^	**9 ^b^**	**−13 ^a^**
A4	The clinician navigates me to take responsibility for my health.	2	6 ^b^	8 ^b^	−9 ^a^
	**Category B: Health Literacy**				
B1	The clinician provides me with specific information for my conditions, not just general info.	−1	−4 ^a^	2 ^b^	0 ^b^
B2	The clinician recommends an accredited site to learn about my disease.	0	−3 ^a^	1 ^b^	2 ^b^
B3	The clinician ensures that I understood the information.	0	−3 ^a^	0 ^b^	3 ^b^
B4	The clinician encourages me to ask questions.	−1	−4 ^a^	−3 ^ab^	4 ^b^
	**Category C: Attentive Listening**				
C1	The clinician invests all the time that I need.	0	−9 ^a^	5 ^b^	5 ^b^
C2	The clinician cares about my feelings.	−1	−7 ^a^	1 ^b^	5 ^c^
C3	The clinician enables me to express myself without interrupting.	1	−6 ^a^	3 ^b^	6 ^b^
C4	The clinician invests more time if I need it.	1	−7 ^a^	5 ^b^	5 ^b^
	**Category D: Non-verbal Communication**				
D1	The clinician respects my time.	2	**10 ^c^**	−10 ^a^	4 ^b^
D2	The clinician does not maintain direct eye contact with me.	1	**9 ^c^**	−13 ^a^	4 ^b^
D3	The clinician is short and succinct.	−1	7 ^c^	−12 ^a^	2 ^b^
D4	Upon entry to the telehealth visit, the clinician greets me warmly.	2	**11 ^c^**	−12 ^a^	6 ^b^

Letters indicate the homogenous subgroups created by Tukey post hoc tests after running an analysis of variance on the individual regression coefficients.

## Data Availability

All relevant data is included in the paper.

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
