# Peer review of "Personalizing Communication of Clinicians with Chronically Ill Elders in Digital Encounters—A Patient-Centered View"

_healthcare, 2024, doi:10.3390/healthcare12040434_

Round 1

Reviewer 1 Report

Comments and Suggestions for Authors

1. I find Abstract not enough informative. I suggest to put this sentence 8from Introduction) instead of second sentence in Abstract: This study explores expectations and preferences of chronically ill elderly patients regarding specific communication messages for communication with clinicians in telemedicine.

2. Line 118: conjoint0-based -I think you should erase 0 in this word.

3. B2 question in Table 1, under Health literacy: please change word  "accredicted" into accredited.

4. For me, Table 2. is not fully understandable. Maybe you should include explanation how the MS groups were detected from these numbers (in the description of Table).

5. Lines 266-282 - maybe you should implement this text into results - otherwise the type of mindset is not clearly described. 

6. In addition to above mentioned - you should explain your results in context of previous researches in the same area.

7. Lines 295-297: To meet patients' expectations and center patients, clinicians are called upon to support communication with patient mindset-tailored messages which consider the most important emphases for members of each mindset. - maybe to say as:

To meet patients' expectations and to put patients into focus, clinicians are advised to support communication with patient using mindset-tailored messages which consider the most important emphases for members of each mindset.

Author Response

We greatly appreciate your time, efforts, and suggestions to improve this manuscript.  Your comments are in Bold and our point-to-point responses in the revision letter and the revised manuscript are underlined.   

Comments and Suggestions for Authors

1. I find the Abstract not enough informative. I suggest putting this sentence (from Introduction) instead of second sentence in Abstract: This study explores expectations and preferences of chronically ill elderly patients regarding specific communication messages for communication with clinicians in telemedicine.

Thank you, to make the abstract more informative, we replaced this sentence in the abstract per your suggestion.

  1. Line 118: conjoint0-based -I think you should erase 0 in this word.

Sorry for this mishap. We omitted the digit from the term conjoint-based. 

  1. B2 question in Table 1, under Health literacy: please change word "accredicted" into accredited.

Again, sorry, we replaced the misspelled word accredicted with the word accredited.

  1. For me, Table 2. is not fully understandable. Maybe you should include explanation how the MS groups were detected from these numbers (in the description of Table).

We added the following explanation to clarify how the mindsets were detected, preceding Table #2:

"Based on K-means mathematical clustering, commonalities in patterns of response of respondents to each message were revealed, facilitating the identification of three mindsets. The Anova Post Hoc test showed that indices indicate that these three mindsets are dignificantly distinct." 

  1. Lines 266-282 - maybe you should implement this text into results - otherwise the type of mindset is not clearly described.

You are right, moving these sentences to the results section makes much more sense. We moved Lines 266 to 282 from the discussion section to the results section.

  1. 6. In addition to above mentioned - you should explain your results in the context of previous research in the same area.

We presented the results in the context of previous studies which supported segmentation of the expectations of patients from communication with clinicians, based on mindsets rather than on demographics.

From the revised manuscript:

"These findings echo previous studies that suggested that patients should be segmented by their expectations from clinicians based on their mindset-belonging rather than on demographics, driving patient satisfaction and health behaviors, and enabling clinicians to use mindset-tailored communication, to meet patient expectations and enhance quality of care" [60-65].

  1. Lines 295-297: To meet patients' expectations and center patients, clinicians are called upon to support communication with patient mindset-tailored messages which consider the most important emphases for members of each mindset. - maybe to say as: To meet patients' expectations and to put patients into focus, clinicians are advised to support communication with patients using mindset-tailored messages which consider the most important emphases for members of each mindset.

We replaced the original text with the wording you suggested.

Dear Reviewer, you marked the results as requiring an improved presentation. 

To clarify, we modified the data analysis and results section as follows.

"2.5 Data Analysis

Ordinary least regression was performed on an individual level, for each message. [51] Ordinary least regression was performed for the whole sample by gender and age group to test the contribution of each message to the importance of the vignette of messages. Next, k-means clustering analysis was performed to uncover the response patterns of respondents to each message with 1-Pearsons’s R distance measure. [49,51,52] This group level analysis of response patterns yielded subgroups (mindsets). ANOVA and Tukey post hoc tests were performed to test if differences among mindsets were significant. [53-55] The high coefficients in each mindset, guided the characterization of what is most important to members of each mindset. To connect the insights from this study to practice we developed a predictive algorithm to easily assign a chronically ill elder patient in the population to a sample mindset.

  1. Results

3.1 Preliminary analysis

We used split halves to test reliability by testing internal consistency between the data for the whole sample and data for half of the sample (0.70; 0.76). Sixty-one percent were females (367) and thirty-nine percent (233) were males. The age ranged from 65 to 92 years. The age distribution amongst the four age groups was as following: 377 respondents age group 65-70; 140 respondents, age group 71-79; 59 respondents, age group 80-87; and 24 respondents, age group 88-95.

3.2 Secondary Analysis

Data analysis was done using R-project (version R-3.6.0) and lm.beta package (R Core Team, 2020). The additive constant is the intercept in the linear regression equation and presents the likelihood that respondents will agree with a vignette of messages. High additive constants (60+) represent groups who strongly agreed with the vignettes and low additive constants (<35) represent groups whose agreement with the vignettes was low. The additive constant was .53 for the whole sample and .55 for the age group of 65-70, reflecting the high importance of the subject. To crystalize the results of the importance of each message, this outcome variable was recoded as a categorical variable. Ratings of 1-6 recoded as zero and ratings of 7-9 were recoded as 100.

The regression analysis yielded coefficients for the 16 messages. The higher the coefficients the higher the importance of the message. In conjoint analysis, coefficients of six and higher are statistically significant indicating a high importance of the message. Coefficients that are lower than 6, or negative coefficients, indicate a lack of agreement with the message. The regression analysis was performed for the whole sample by gender and age testing the contribution of each message to the importance of the whole vignette of messages [51]. With 600 respondents and 24 vignettes rated by each respondent, there were 600 (14,000) different observations for the regression analysis (Systat Software Inc, SYSTAT v13.2. 2020.www.sigmaplot.com). Regression coefficients for the total sample were not significant, indicating that there were no significant differences in the importance of messages in communication via telemedicine.

Likewise, no significant differences were found between the coefficients of males and females and across age groups. Looking into differences among age groups, the strongest expectation of respondents in the age group of 80-87 was "The Clinician greets me warmly when he sees me” (beta =10). The strongest expectations of respondents in the age groups of 88-95 were: “The Clinician provides me with information that is specific to my conditions, not just general info" (beta =11) and "The Clinician guides me through a process to maintain my health" (beta =12).

Group (Binary Ratings) Total

MS1

MS2

MS3

Base Size

600

217

194

189

Additive Constant

53

54

53

51

Category A: Internal Locus of Control

A1

The Clinician walks me through a process of change to maintain my health.

1

6b

9b

-12a

A2

The clinician initiates a dialogue about my expectations.

1

6b

12b

-14a

A3

The clinician explains the treatment plan.

1

7b

9b

-13a

A4

The clinician navigates me to take responsibility for my health.

2

6b

8b

-9a

Category B: Health Literacy

B1

The clinician provides me with specific information for my conditions, not just general info.

-1

-4a

2b

0b

B2

The clinician recommends an accredited site to learn about my disease.

0

-3a

1b

2b

B3

The Clinician ensures that I understood the information.

0

-3a

0b

3b

B4

The clinician encourages me to ask questions.

-1

-4a

-3ab

4b

Category C:  Attentive Listening

C1

The clinician invests all the time that I need.

0

-9a

5b

5b

C2

The clinician cares about my feelings.

-1

-7a

1b

5c

C3

The clinician enables me to express myself without interrupting.

1

-6a

3b

6b

C4

The clinician invests more time if I need it.

1

-7a

5b

5b

Category D:  Non-verbal Communication

D1

The clinician respects my time.

2

10c

-10a

4b

D2

The clinician does not maintain direct eye contact with me.

1

9c

-13a

4b

D3

The clinician is short and succinct.

-1

7c

-12a

2b

D4

Upon entry to the telehealth visit, the clinician greets me warmly.

2

11c

-12a

6b

Next, using k-means mathematical clustering, we analyzed response patterns to each message with 1-Pearsons’s R distance measure. [49,51,52] Commonalities in response patterns to each message uncovered groups of participants with similar responses to messages. [53] The clustering analysis yielded three subgroups (mindsets). ANOVA and Tukey post hoc tests showed indices that indicated that the differences among mindsets were significant. [55] The three mindsets are based on the mathematical clustering, from which the commonalities in patterns of response to each message, facilitated the assignment of each participant into mindsets presenting distinct expectations of elder patients from communication with clinicians in Telemedicine. Mindsets reflect various ways of thinking about communication with clinicians. In each mindset, the highest coefficients guided the characterization of the expectations of members in that mindset [56, 57]. Table 2 presents mindset segmentation. Figure 1 illustrates the mathematical clustering of messages by importance.

Table 2: Results of Segmentation by Mindset (MS) and Tukey post hoc tests

Letters indicate the homogenous subgroups created by Tukey post hoc tests after running analysis of variance on the individual regression coefficients.

Figure 1. An example of mathematical clustering to identify commonalities by perceived importance of messages.

Members of Mindset 1, comprising 36% of the sample, were most concerned with non-verbal communication; they sought a warm greeting and eye contact. The growing elderly population and the immense benefits of telemedicine call upon clinicians to ad-dress needs of chronically ill patients in telemedicine communication and optimize its utilization.[56] Members of mindset 2, comprising 32% of the sample, are interested in communication that enhances internal locus of control. They would like to talk about their expectations and for the clinician to guide them through a change process to take responsibility and maintain their health. They would like relevant specific information about their condition rather than general information about the disease. The strongest messages for this mindset were "The clinician walks me through a change process to maintain my health: (beta= 12); "The clinician talks to me about my expectations" (beta=12) and "The clinician helps me take responsibility for my health" (beta=9). They expect clinicians to use messages that empower and navigate them to better manage their illness leading to higher resilience. [60–62] Members of mindset 3, comprising 31% of the sample, have an external locus of control and strongly oppose a change process guided by the Clinician to maintain their health (beta= -12) or to help them take responsibility for their health (beta=- 13) or dialogue about their communication expectations (beta= -14).

To connect the insights from this study to practice, we developed a predictive algorithm. To develop the algorithm, we considered a total of sixty-four potential response patterns to each message and the messages that strongly distinguished among the three mindsets. Based on six messages that distinguished among the mindsets, we employed a mathematical Monte Carlo simulation and developed the predictive algorithm [56]. The predictive algorithm enables clinicians to may easily assign a chronically ill elder patient in the clinic to one of the study mindsets and choose to communicate by mindset-tailored communication messages. Since respondents are distributed across age groups and genders in the population, we created a predictive algorithm to quickly assign elderly patients to mindsets and accord communication to meet the expectations of members of each mindset. Figure 2 presents the predictive algorithm available at https://www.pvi360.com/TypingToolPage.aspx?projectid=2312&userid=2008

Figure 2. The predictive algorithm lists the six most discriminating elements with binary answers needed to predict the mindset memberships of new participants."

Thank you again for your thorough work. We hope we adequately responded to your comments.

All the best and a healthy 2024

Reviewer 2 Report

Comments and Suggestions for Authors

This paper studied chronically ill elderly patients and clinicians in telehealth encounters, proposing a predictive algorithm for personalized communication to enhance patient-centered care.

This topic is important for related readers in this field. The methodology is sound and the results can support the conclusions.

I would suggest to further discuss the generalization ability of the statistical models.

Consider incorporating more visual graphs to illustrate crucial data points and trends. This enhancement will not only elevate the presentation but also render the information more accessible to a wider audience.

Author Response

We thank the editor and the reviewers and greatly appreciate your time, efforts, and suggestions to improve this manuscript.  Comments are in Bold and our point-to-point response in the revision letter and the revised manuscript are underlined.   

This paper studied chronically ill elderly patients and clinicians in telehealth encounters, proposing a predictive algorithm for personalized communication to enhance patient-centered care. This topic is important for related readers in this field. The methodology is sound, and the results can support the conclusions.

Thank you.

I would suggest further discussing the generalization ability of the statistical models.

Thank you. This is an important point. We added a limitations section to the discussion where we discussed the limitations. From the revised manuscript:

"This study on a novel strategy to test communication expectations of chronically ill elders in Telemedicine is not without limitations. First, while the presented novel statistical strategy itself is replicable and generalizable across health settings and countries, and is expected to yield distinct mindsets of patients, communication has cultural roots which reduce the generalizability of the communication messages themselves, by mindsets. First, attitudes towards the elderly differ between Western and Eastern cultures, whereas in the latter, the elderly are highly respected, and all their needs must be fulfilled affecting patient-clinician communication in Telemedicine. Second, there are cultural differences in attitudes and perceptions of authority figures that may affect the involvement of elders in communication with clinicians as authority figures, regardless of whether or not the clinician will use mindset-tailored communication messages. Third, our global landscape is becoming rich in immigrant diversity and elderly immigrants may not be fully versed in the dominant local language. Another limitation is that we didn't test elder patient expectations by diseases included in the study, which may yield different results of mindset-belonging and varying importance levels of the messages in each mindset.  Along with the patient-centered care framework, future studies may test the expectations of chronically ill patients from clinicians, by disease and by culture.  Also, future studies may test the effect of implementing mindset-tailored messages with chronically ill elder patients on their satisfaction with interactions with clinicians in telemedicine."

Consider incorporating more visual graphs to illustrate crucial data points and trends. This enhancement will not only elevate the presentation but also render the information more accessible to a wider audience.

We incorporated Figure 1 As an example of the mathematical clustering used to identify commonalities in response patterns of respondents to each message by ratings of importance.

Figure 1. An example of mathematical clustering performed to identify commonalities by the perceived importance of each message.

Thank you again for your thorough work. We hope we adequately responded to your comments.

All the best and a healthy 2024

Reviewer 3 Report

Comments and Suggestions for Authors

Thank you for the opportunity to review the manuscript. Digital health and patient-centered view are relevant recommendations to mitigate the global pressure of chronically ill elderly patients.

I would like to leave some suggestions and questions for the authors.

- Lines 10-12: “Chronically ill elderly patients are concerned about losing the personal connection with clinicians in digital encounters and clinicians are concerned about missing medical information jeopardizing quality of care. This experiment addresses this issue.” However, in lines 85-87, we can read: “This study explores expectations and preferences of chronically ill 85 elderly patients regarding specific communication messages for communication with clinicians in telemedicine…”

Did this study address missing medical information jeopardizing quality of care?  Please rewrite the objectives.

 -Lines 85, 90 and 258: The application of the questionnaire, as performed in this study, is a quantitative study. The word "explore" is best used for qualitative studies. I suggest replacing the word "explore" with the verb "identify" or "verify".

-Lines 100-101: “The sample comprised of 600 chronically ill elderly patients who received medical care through digital encounters since the outbreak of the COVID-19 pandemic…”

·   What were the inclusion and exclusion criteria for chronically ill elderly participants?

·     Aligning with “Patient-Centered View”, the patient's pathologies should be highlighted. What diseases did these elderly people have? Cardiovascular disease? Neurological disease? Psychiatric illness? Which? Depending on the disease, there may be a lesser or greater need for non-verbal communication, for example. This may cause a bias in the results.

-Line 232: Please describe “PVI” in the text.

-Lines 266-282: Please this information should be described in the "Results" section.

-Lines 284-291: Please this information should be described in the "Discussion" section.

-The discussion can be further developed, answering the research questions (What messages do elderly chronically ill patients prefer in communication in telemedicine? Do different subgroups of chronically ill patients differ in their preferences for telemedicine communication with Clinicians?)

-The authors did not describe the limitations of the study.

-Lines 292-302: Please this information should be described in the "Conclusion" section.

Author Response

Dear Reviewer,

We greatly appreciate your time, efforts, and suggestions to improve this manuscript.  Comments are in Bold and our point-to-point response in the revision letter and the revised manuscript are underlined.   

Digital health and patient-centered view are relevant recommendations to mitigate the global pressure of chronically ill elderly patients. I would like to leave some suggestions and questions for the authors.

Thank you.

Lines 10-12: “Chronically ill elderly patients are concerned about losing the personal connection with clinicians in digital encounters and clinicians are concerned about missing medical information jeopardizing quality of care. This experiment addresses this issue.” However, in lines 85-87, we can read: “This study explores expectations and preferences of chronically ill 85 elderly patients regarding specific communication messages for communication with clinicians in telemedicine…”  Did this study address missing medical information jeopardizing quality of care?  Please rewrite the objectives. 

Thank you. The barriers of using telemedicine, among patients and clinicians described in Lines 10-12 are based on the literature. Clinicians view telemedicine as entailing potential risks of missing information (cues) for the diagnosis. The study objectives appear in lines 85-87. To clarify, we changed the wording in lines 10-12 to the following:

"Clinicians are concerned about missing nonverbal cues that are important for the diagnosis jeopardizing quality of care."

Lines 85, 90 and 258: The application of the questionnaire, as performed in this study, is a quantitative study. The word "explore" is best used for qualitative studies. I suggest replacing the word "explore" with the verb "identify" or "verify."

Thank you. We replaced the term explore with the term validate.

Lines 100-101: “The sample comprised of 600 chronically ill elderly patients who received medical care through digital encounters since the outbreak of the COVID-19 pandemic…”

What were the inclusion and exclusion criteria for chronically ill elderly participants?

Inclusion criteria were chronic illnesses including Cardiovascular disease, Neurological disease, Diabetes, Metabolic disorders, Lung disease, Chronic back pain, Osteoporosis. Exclusion criteria were mental illnesses and Cancer which differ in their constructs.

Aligning with “Patient-Centered View”, the patient's pathologies should be highlighted. What diseases did these elderly people have? Cardiovascular disease? Neurological disease? Psychiatric illness? Which? Depending on the disease, there may be a lesser or greater need for non-verbal communication, for example. This may cause a bias in the results.

This is an important comment. Since we didn't test expectations of elders by disease, under the inclusion criteria, we added this as a limitation in the limitations paragraph that we added in the discussion section. WE also included it as a direction for future research.   

From the revised manuscript:

" Another limitation is that we didn't test elder patient expectations by diseases included in the study, which may yield different results of mindset-belonging and varying importance level of the messages in each mindset.  Along with the patient-centered care framework, future studies may test the expectations of chronically ill patients from clinicians, by disease and by cultures.  Also, future studies may test the effect of implementing mindset-tailored messages with chronically ill elder patients on their satisfaction with interactions with clinicians in telemedicine."

Line 232: Please describe “PVI” in the text.

We replaced the term PVI with the term predictive algorithm.

Lines 266-282: Please this information should be described in the "Results" section.

This paragraph was moved to the Results section.

Lines 284-291: Please this information should be described in the "Discussion" section.

We moved it to the discussion section.

The discussion can be further developed, answering the research questions (What messages do elderly chronically ill patients prefer in communication in telemedicine? Do different subgroups of chronically ill patients differ in their preferences for telemedicine communication with Clinicians?

Thank you. We elaborated on these questions in the discussion section. From the revised manuscript:

"What messages do elderly chronically ill patients prefer in communication in telemedicine?

 Surprisingly, no significant differences were found between male and female chronically ill elders across age groups. The strongest expectation of elders in the age group of 80-87 was that the clinicians will greet them warmly upon the beginning of the virtual encounter. The strongest expectations of elders in the age group of 88-95 were that the clinicians will provide them with information that is specific to their conditions, rather than general information and, that clinicians will guide them through a process to maintain their health.

Do different subgroups of chronically ill patients differ in their preferences for telemedicine communication with Clinicians?

Three mindsets which are similar in size. represent subgroups of chronically ill patients and the expectations of members of one mindset are significantly distinct from expectations of members of other mindsets. Thus, expectations of members of one MS are irrelevant to members of other mindsets. Members of Mindset 1, expect a warm greeting and eye contact. Members of Mindset 2, expect communication that enhances their internal locus of control, a dialogue concerning their expectations and for clinicians to guide them through a change process to maintain their health. These elders expect specific information about their condition and for the clinician to help them take responsibility for their health, empower them, and navigate them to better manage their illness leading to higher resilience. Members of Mindset 3 have an external locus of control and oppose a dialogue and a change process guided by clinicians."

The authors did not describe the limitations of the study.

True. We added a paragraph on limitations in the discussion section. From the revised manuscript:

"This study on a novel strategy testing communication expectation of chronically ill elders in Telemedicine is not without limitations. First, while the presented novel statistical strategy itself is replicable and generalizable across health settings and countries, and is expected to yield distinct mindsets of patients, communication has cultural roots which reduce the generalizability of the communication messages themselves, by mindsets. First, attitudes towards the elderly differ between Western and Eastern cultures, where in the latter, the elderly are highly respected, and all their needs must be fulfilled affecting patient-clinician communication in Telemedicine. Second, there are cultural differences in attitudes and perceptions of authority figures that may affect the involvement of elders in communication with clinicians as authority figures, regardless of whether or not the clinician will use mindset-tailored communication messages. Third, our global landscape is becoming rich in immigrant diversity and elderly immigrants may not be fully versed in the dominant local language. Another limitation is that we didn't test elder patient expectations by diseases included in the study, which may yield different results of mindset-belonging and varying importance level of the messages in each mindset.  Along with the patient-centered care framework, future studies may test the expectations of chronically ill patients from clinicians, by disease and by cultures.  Also, future studies may test the effect of implementing mindset-tailored messages with chronically ill elder patients on their satisfaction with interactions with clinicians in telemedicine."

Lines 292-302: Please this information should be described in the "Conclusion" section.

Thanks. We moved it to the conclusion section.

Dear Reviewer, since you marked the design, methods, and results as requiring some improvement ("can be improved"), we simplified the data analysis, and the results as follows. From the revised manuscript:  

To clarify, we modified the data analysis and results section as follows. From the revised manuscript

"2.5 Data Analysis

Ordinary least regression was performed on an individual level, for each message. [51] Ordinary least regression was performed for the whole sample by gender and age group to test the contribution of each message to the importance of the vignette of messages. Next, k-means clustering analysis was performed to uncover the response patterns of respondents to each message with 1-Pearsons’s R distance measure. [49,51,52] This group level analysis of response patterns yielded subgroups (mindsets). ANOVA and Tukey post hoc tests were performed to test if differences among mindsets were significant. [53-55] The high coefficients in each mindset, guided the characterization of what is most important to members of each mindset. To connect the insights from this study to practice we developed a predictive algorithm to easily assign a chronically ill elder patient in the population to a sample mindset.

  1. Results

3.1 Preliminary analysis

We used split halves to test reliability by testing internal consistency between the data for the whole sample and data for half of the sample (0.70; 0.76). Sixty-one percent were females (367) and thirty-nine percent (233) were males. The age ranged from 65 to 92 years. The age distribution amongst the four age groups was as following: 377 respondents age group 65-70; 140 respondents, age group 71-79; 59 respondents, age group 80-87; and 24 respondents, age group 88-95.

3.2 Secondary Analysis

Data analysis was done using R-project (version R-3.6.0) and lm.beta package (R Core Team, 2020). The additive constant is the intercept in the linear regression equation and presents the likelihood that respondents will agree with a vignette of messages. High additive constants (60+) represent groups who strongly agreed with the vignettes and low additive constants (<35) represent groups whose agreement with the vignettes was low. The additive constant was .53 for the whole sample and .55 for the age group of 65-70, reflecting the high importance of the subject. To crystalize the results of the importance of each message, this outcome variable was recoded as a categorical variable. Ratings of 1-6 recoded as zero and ratings of 7-9 were recoded as 100.

The regression analysis yielded coefficients for the 16 messages. The higher the coefficients the higher the importance of the message. In conjoint analysis coefficients of six and higher are statistically significant indicating a high importance of the message. Coefficients that are lower than 6, or negative coefficients, indicate a lack of agreement with the message. The regression analysis was performed for the whole sample by gender and age testing the contribution of each message to the importance of the whole vignette of messages [51]. With 600 respondents and 24 vignettes rated by each respondent, there were 600 (14,000) different observations for the regression analysis (Systat Software Inc, SYSTAT v13.2. 2020.www.sigmaplot.com). Regression coefficients for the total sample were not significant, indicating that there were no significant differences in the importance of messages in communication via telemedicine.

Likewise, no significant differences were found between the coefficients of males and females and across age groups. Looking into differences among age groups, the strongest expectation of respondents in the age group of 80-87 was "The Clinician greets me warmly when he sees me” (beta =10). The strongest expectations of respondents in the age groups of 88-95 were: “The Clinician provides me with information that is specific to my conditions, not just general info" (beta =11) and "The Clinician guides me through a process to maintain my health" (beta =12).

Next, using k-means mathematical clustering, we analyzed response patterns to each message with 1-Pearsons’s R distance measure. [49,51,52] Commonalities in response patterns to each message uncovered groups of participants with similar responses to messages. [53] The clustering analysis yielded three subgroups (mindsets). ANOVA and Tukey post hoc tests showed indices that indicated that the differences among mindsets were significant. [55] The three mindsets are based on the mathematical clustering, from which the commonalities in patterns of response to each message, facilitated the assignment of each participant into mindsets presenting distinct expectations of elder patients from communication with clinicians in Telemedicine. Mindsets reflect various ways of thinking about communication with clinicians. In each mindset, the highest coefficients guided the characterization of the expectations of members in that mindset [56, 57]. Table 2 presents mindset segmentation. Figure 1 illustrates the mathematical clustering of messages by importance.

Members of Mindset 1, comprising 36% of the sample, were most concerned with non-verbal communication; they sought a warm greeting and eye contact. The growing elderly population and the immense benefits of telemedicine call upon clinicians to address the needs of chronically ill patients in telemedicine communication and optimize its utilization.[56] Members of Mindset 2, comprising 32% of the sample, are interested in communication that enhances internal locus of control. They would like to talk about their expectations and for the clinician to guide them through a change process to take responsibility and maintain their health. They would like relevant specific information about their condition rather than general information about the disease. The strongest messages for this mindset were "The clinician walks me through a change process to maintain my health: (beta= 12); "The clinician talks to me about my expectations" (beta=12) and "The clinician helps me take responsibility for my health" (beta=9). They expect clinicians to use messages that empower and navigate them to better manage their illness leading to higher resilience. [60–62] Members of mindset 3, comprising 31% of the sample, have an external locus of control and strongly oppose a change process guided by the Clinician to maintain their health (beta= -12) or to help them take responsibility for their health (beta=- 13) or a dialogue about their communication expectations (beta= -14).

To connect the insights from this study to practice, we developed a predictive algorithm. To develop the algorithm, we considered the total of sixty-four potential response patterns to each message and the messages that strongly distinguished among the three mindsets. Based on six messages that distinguished among the mindsets, we employed a mathematical Monte-Carlo simulation, and developed the predictive algorithm [56]. The predictive algorithm enables clinicians to may easily assign a chronically ill elder patient in the clinic to one of the study mindsets and choose to communicate by mindset-tailored communication messages. Since respondents are distributed across age groups and genders in the population, we created a predictive algorithm to quickly assign elderly patients into mindsets and accord communication to meet the expectations of members of each mindset. Figure 2 presents the predictive algorithm available at https://www.pvi360.com/TypingToolPage.aspx?projectid=2312&userid=2008

Figure 2. The predictive algorithm lists the six most discriminating elements with binary answers needed to predict the mindset membership."

You also marked the conclusions section as needing improvement. We revised the section as follows:

  1. Conclusions

In terms of practice implications, the insights developed from this study enable clinicians to employ our predictive algorithm or develop one, to easily assign elder patients with chronic illnesses (excluding mental illness) in telemedicine, into mindsets from the sample. Detecting the mindset to which a patient belongs and understanding the strongest expectations of members in that mindset, will provide clinicians with the choice to communicate with the patient through personalized, mindset-tailored messages. The greater personalization will remove a major barrier of patients regarding telemedicine. Moreover, since personalized communication was proven to establish higher patient trust in clinicians, it may yield trust in clinicians in telemedicine as well.  Trust was proven to enhance medication- adherence, and self-management of diseases [66]. Hence, it is possible that using mindset-tailored communication with chronically ill elder patients will contribute to optimal self-management of chronic diseases, creating added value to telemedicine and enhancing the quality of care. [67-72] Patient greater utilization of telemedicine and patient involvement may increase if clinicians meet patient expectations in communication with clinicians which may improve health behaviors and overall clinical outcomes.[28,69].

Thank you again for your thorough work. We hope we adequately responded to your comments.  

All the best and a healthy 2024

Round 2

Reviewer 1 Report

Comments and Suggestions for Authors

Thank you for all corrections and clarifications.

Author Response

Thank you for your time. 

Best,

The authors 

Reviewer 2 Report

Comments and Suggestions for Authors

The authors have addressed my previous concerns and I think this vesrion is ready to be accepted.

Author Response

Thanks for your time.

The authors

Reviewer 3 Report

Comments and Suggestions for Authors

Congratulations to the authors for the manuscript. I would like to make a few comments.

 -Lines 109-112: “ Inclusion criteria were chronic illnesses such as Cardiovascular disease, 109 Neurological disease, Diabetes, Metabolic disorders, Lung disease, Chronic back pain, Osteoporosis. Exclusion criteria was mental illness and Cancer which differ in their constructs.”

·         Exclusion criteria was mental illness and Cancer   OR    Exclusion criteria WERE mental illness and Cancer? 

·         Since this information reads to the elderly, this text could be written after line 106, like this: 

“The sample comprised of 600 chronically ill elderly patients who received medical care through digital encounters since the outbreak of the COVID-19 pandemic and accords with the suggested number of respondents in studies that aim at stable utilities [49]. The inclusion criteria were chronic illnesses such as Cardiovascular disease, Neurological disease, Diabetes, Metabolic disorders, Lung disease, Chronic back pain, Osteoporosis. Exclusion criteria WERE mental illness and Cancer which differ in their constructs. Six hundred undergraduate students in business administration programs at one of four campuses of the academic institution with which the second author is affiliated interviewed patients…”

Figure 2.

-Mind-set   OR   mindset?

- Please describe PVI which is in the picture (example: (Legend: PVI= Personal Viewpoint Identifier)

Discussion

-Please the word "study" is repeated in the first two paragraphs.

- The research questions seem loose in the text. How about writing a few sentences before the questions? For example:

This work allowed to answer the following research questions:

aa) What messages do elderly chronically ill patients prefer in communication in telemedicine?  Surprisingly, no significant differences were found between male and female chronically ill elders across age groups. The strongest expectation of elders ages 80-87 was …   

b) Do different subgroups of chronically ill patients differ in their preferences for telemedicine com munication with Clinicians? Three mindsets which are similar in size. represent subgroups of chronically ill patients  and the expectations of members of …

-Line 296: “Three mindsets which are similar in size. represent subgroups of chronically ill patients…”   OR  Three mindsets, which are similar in size, represent subgroups of chronically ill patients?

-Line 324: Is the word "First" right here?

CONCLUSIONS

In the "Conclusion" section, no bibliographic references are placed. These must be mentioned before.

Lines 361-371:  This section could have been written before the “Conclusion” section. 

Author Response

Thank you for your time and suggestions to improve this manuscript. Your comments are in Bold and point-to-point responses are underlined both below and in the revised manuscript.    

Lines 109-112: “Inclusion criteria were chronic illnesses such as cardiovascular diseases, 109 Neurological disease, Diabetes, Metabolic disorders, Lung disease, Chronic back pain, Osteoporosis. Exclusion criteria was mental illness and Cancer which differ in their constructs.”

  • Exclusion criteria was mental illness and Cancer   OR    Exclusion criteria WERE mental illness and Cancer?

       Yes, thank you. I corrected the spelling error and replaced 'was' with 'were.'

  • Since this information reads to the elderly, this text could be written after line 106, like this: 

“The sample comprised of 600 chronically ill elderly patients who received medical care through digital encounters since the outbreak of the COVID-19 pandemic and accords with the suggested number of respondents in studies that aim at stable utilities [49]. The inclusion criteria were chronic illnesses such as Cardiovascular diseases, Neurological diseases, Diabetes, Metabolic disorders, Lung disease, Chronic back pain, Osteoporosis. Exclusion criteria were mental illness and Cancer which differ in their constructs. Six hundred undergraduate students in business administration programs at one of four campuses of the academic institution with which the second author is affiliated interviewed patients…”

Thank you for this comment. I moved the above paragraph to line 106. It does read better now.  

Figure 2.

-Mind-set   OR   mindset?

Please describe PVI which is in the picture (example: (Legend: PVI= Personal Viewpoint Identifier)

Good spotting!! The Legend for figure 2 was modified. The title is now:

"The predictive algorithm listing the six most discriminating elements with binary answers needed to predict the mindset-membership of new participants."

The "PVI" was replaced with The predictive algorithm.  

Discussion

-Please the word "study" is repeated in the first two paragraphs.

The first two paragraphs were revised as follows:

This study tested expectations of elderly chronically ill patients from communication messages in telemedicine presenting a novel strategy of testing patterns of thinking and expectations of elders from communication in telemedicine, creating a predictive algorithm for detecting patient mindset and using mindset-tailored communication with clinicians. The identified mindsets contradict previous findings which viewed patient expectations from telehealth services as varying by services provided and respondent demographics [58]. The study filled several gaps in the knowledge. First, the study identified what elders with chronic illness expect and perceive as important in communication with clinicians in telemedicine and how they are segmented in their expectations. Methodologically, this study conducted a conjoint-based experimental design that reduces biases of typical observational studies while testing a great many messages with no limitations of degrees of freedom.

- The research questions seem loose in the text. How about writing a few sentences before the questions? For example:

This work allowed to answer the following research questions:

  1. a)What messages do elderly chronically ill patients prefer in communication in telemedicine?  Surprisingly, no significant differences were found between male and female chronically ill elders across age groups. The strongest expectation of elders ages 80-87 was …   
  2. b) Do different subgroups of chronically ill patients differ in their preferences for telemedicine communication with Clinicians? Three mindsets which are similar in size. represent subgroups of chronically ill patients and the expectations of members of …

Line 296: “Three mindsets which are similar in size. represent subgroups of chronically ill patients…”   OR  Three mindsets, which are similar in size, represent subgroups of chronically ill patients?

Thanks. The text and the punctuation were revised per your suggestions as follows:

"This work allowed to answer the following research questions:

a). What messages do elderly chronically ill patients prefer in communication in telemedicine?  Surprisingly, no significant differences were found between male and female chronically ill elders across age groups. The strongest expectation of elders ages 80-87 was that the clinicians will greet them warmly upon the beginning of the virtual encounter. The strongest expectations of elders ages 88-95 were that the clinicians will provide them with information that is specific to their conditions, rather than general information and that clinicians will guides them through a process to maintain my health.

b). Do different subgroups of chronically ill patients differ in their preferences for telemedicine communication with Clinicians? Three mindsets, which are similar in size, represent subgroups of chronically ill patients and the expectations of members of one mindset are significantly distinct from expectations of members of other mindsets. Thus, expectations of members of one mindset are irrelevant to members of other mindsets. Members of Mindset 1, expect a warm greeting and eye contact. Members of Mindset 2, expect communication that enhances their internal locus of control, a dialogue concerning their expectations and for clinicians to guide them through a change process to maintain their health. These elders expect specific information about their condition and for the clinician to help them take responsibility for their health, empower them, and navigate them to better manage their illness leading to higher resilience. Members of Mindset 3 have an external locus of control and oppose a dialogue and a change process guided by clinicians."

 -Line 324: Is the word "First" right here?

It is the first limitation of the study. However, to clarify, we revised the sentence as follows:

" First, the generalizability of the study."

CONCLUSIONS

In the "Conclusion" section, no bibliographic references are placed. These must be mentioned before.

Lines 361-371:  This section could have been written before the “Conclusion” section. 

This section from line 361 to line 371 was placed before the conclusions and references which appeared in the conclusion section were placed in the section on clinical implications as follows:

"Telemedicine is developing, expanding, and creating new opportunities to meet patient expectations, to improve quality and delivery of care by personalizing care and closing the chasm between aspiring for PCC and obtaining PCC. To meet patients' expectations and to put patients into focus, clinicians are advised to support communication with patient using mindset-tailored messages which consider the most important emphases for members of each mindset. The use of the predictive algorithm developed for this study detects the mindset-belonging of each elder patient at the clinic linking it to a sample mindset, enabling personalized communication while delivering care via telemedicine. Understanding expectations of patients from communication with clinicians by mindset can structure the telehealth visit more effectively, enhance patient-centered care and quality of care. Moreover, the greater personalization will remove a major barrier of patients regarding telemedicine. Since personalized communication was proven to establish higher patient trust in clinician, it may yield trust in clinicians in telemedicine as well. Trust was proven to enhance medication- adherence and self-management of diseases [66]. Hence, it is possible that using mindset- tailored communication with chronically ill elder patients will contribute to optimal self-management of chronic diseases, creating an added value to telemedicine and enhancing quality of care. [67-72] Patient greater utilization of telemedicine and patient involvement may increase if clinicians meet patient expectations in communication with clinicians which may improve health behaviors and overall clinical outcomes [28,69]."

Thanks for your impressive thoroughness.

Best, The authors.  
